# A Novel 13q12 Microdeletion Associated with Familial Syndromic Corneal Opacification

**DOI:** 10.3390/genes14051034

**Published:** 2023-05-01

**Authors:** Jasmine Y. Serpen, William Presley, Adelyn Beil, Stephen T. Armenti, Kayla Johnson, Shahzad I. Mian, Jeffrey W. Innis, Lev Prasov

**Affiliations:** 1Department of Ophthalmology and Visual Sciences, Kellogg Eye Center, University of Michigan, Ann Arbor, MI 48105, USA; 2Case Western Reserve University School of Medicine, Cleveland, OH 44106, USA; 3Department of Human Genetics, University of Michigan, Ann Arbor, MI 48109, USA; 4Department of Pediatrics, University of Michigan, Ann Arbor, MI 48109, USA; 5Department of Ophthalmology, University of Pennsylvania, Philadelphia, PA 19104, USA

**Keywords:** 13q12.11 microdeletion, corneal opacification, tracheomalacia, laryngomalacia, hearing loss, *XPO4*, *LATS2*, *ZDHHC20*, *IFT88*, SMAD signaling

## Abstract

Progressive corneal opacification can result from multiple etiologies, including corneal dystrophies or systemic and genetic diseases. We describe a novel syndrome featuring progressive epithelial and anterior stromal opacification in a brother and sister and their mildly affected father, with all three family members having sensorineural hearing loss and two also with tracheomalacia/laryngomalacia. All carried a 1.2 Mb deletion at chromosome 13q12.11, with no other noteworthy co-segregating variants identified on clinical exome or chromosomal microarray. RNAseq analysis from an affected corneal epithelial sample from the proband’s brother revealed downregulation of *XPO4*, *IFT88*, *ZDHHC20*, *LATS2*, *SAP18*, and *EEF1AKMT1* within the microdeletion interval, with no notable effect on the expression of nearby genes. Pathway analysis showed upregulation of collagen metabolism and extracellular matrix (ECM) formation/maintenance, with no significantly down-regulated pathways. Analysis of overlapping deletions/variants demonstrated that deleterious variants in *XPO4* were found in patients with laryngomalacia and sensorineural hearing loss, with the latter phenotype also being a feature of variants in the partially overlapping *DFNB*1 locus, yet none of these had reported corneal phenotypes. Together, these data define a novel microdeletion-associated syndromic progressive corneal opacification and suggest that a combination of genes within the microdeletion may contribute to ECM dysregulation leading to pathogenesis.

## 1. Introduction

Genetic disorders, including corneal dystrophies and multisystem conditions with corneal involvement, have significant implications for patients’ visual acuity. Corneal dystrophies are rare (~9/10,000) abnormalities of one or several layers of the cornea and have a genetic or hereditary epigenetic origin [1]. They present with variably shaped corneal opacities with a spectrum of effects on visual acuity and are typically bilateral and progressive [2]. Symptoms may include visual impairment and eye pain from recurrent corneal erosions or disruptions of the corneal epithelium. They can be inherited in an autosomal dominant, autosomal recessive, or X-linked fashion [2]. Traditionally, corneal dystrophies have been classified based on the layer of the cornea affected (epithelial and subepithelial, Bowman layer, stromal, and those affecting the Descemet membrane and the endothelium). However, the International Committee for Classification of Corneal Dystrophies was developed to incorporate genetic information into the anatomic classification system [3]. A category number from 1 to 4 is assigned to reflect the degree to which genetic evidence like chromosomal locus and specific mutations has been determined in relation to the dystrophy [3]. While many corneal dystrophies have defined genetic etiologies (i.e., congenital hereditary endothelial dystrophy, posterior polymorphous corneal dystrophy, and Fuchs’ endothelial corneal dystrophy) [4], for other corneal dystrophies, the genetic etiology has not been identified [5]. Variants in multiple genes may contribute to a single phenotype, and different variants in a single gene can cause different phenotypes [5]. Many anterior corneal dystrophies, i.e., granular and lattice dystrophy, result from dominant variants in *TGFBI* [6]. Within the cornea, the TGFBI protein is mainly expressed in the epithelium and has been posited to be involved in corneal wound healing and extracellular matrix maintenance [7]. Mutated TGFBI protein appears to specifically accumulate in the cornea versus other tissues in patients with TGFBI-associated corneal dystrophies. In these dystrophies, corneal deposits localize across layers of the cornea, including the Bowman layer and the corneal stroma [7].

Corneal involvement is associated with a variety of multisystemic disorders. For example, gelsolin amyloidosis is a multisystem disorder clinically associated with corneal amyloidosis that resembles lattice corneal dystrophy [8]. Keratopathy is also present in inborn errors of metabolism, including mucopolysaccharidoses, mucolipidoses, cystinosis, tyrosinemia type II, Wilson disease, Fabry disease, Lecithin Cholesterol Acyltransferase Deficiency (LCAT)-related metabolic disease, and Tangier disease [8]. Corneal opacification is a feature of various systemic conditions, including ectodermal dysplasias, of which hundreds of clinically defined subtypes exist [8]. Ectodermal dysplasias affect ectoderm-derived ocular structures, including the corneal epithelium. For instance, ectrodactyly-ectodermal-dysplasia-clefting syndrome includes ocular features of limbal stem cell deficiency, corneal scarring, and ocular inflammation. Keratitis-ichthyosis-deafness syndrome can include neovascularization, recurrent corneal epithelial defects, and limbal stem cell deficiency [8]. Limbal stem cell deficiency is a reported feature of various genetic disorders in addition to ectrodactyly-ectodermal-dysplasia-clefting syndrome and keratitis-ichthyosis-deafness syndrome, including PAX6-related aniridia, xeroderma pigmentosum, and Turner syndrome [9,10,11,12].

Microdeletion syndromes can be associated with various systemic and ocular features by affecting multiple genes. Various 13q12.11 microdeletions have been reported in the literature to date, with features including global developmental delay, intellectual disability, hearing impairment, laryngomalacia, atrial septal defect, cryptorchidism, delayed speech and language development, microcephaly, myopia, and hypotonia [13,14,15,16]. Here, we describe a novel familial syndrome associated with progressive epithelial/anterior stromal corneal opacification and explore its association with a 13q12.11 microdeletion. We use gene expression analysis of a patient’s corneal epithelial sample to inform disease pathogenesis.

## 2. Materials and Methods

### 2.1. Clinical Testing

Chromosomal microarray and karyotype testing were performed on the proband, with confirmatory testing on her brother, father, and unaffected mother using standard clinical-based testing with the Nimblechip HG18 385,000 probe array (Roche, Madison, WI, USA). DNA samples were also collected from these 4 individuals with OraCollect buccal swabs and submitted for clinical whole exome sequencing (WES) using the XomeDxPlus platform (GeneDx, Stamford, CT, USA) and processed using the standard GeneDx clinical pipeline, including the GeneDx proprietary capture kit (NGS-CNV), paired-end sequencing on the Illumina platform, alignment with genome build GRCh37/UCSC hg19 and XomeAnalyzer for variant calling (including CNVs of greater than 3 exons) and filtering. The mean depth of coverage was 121×, with 98.9% of the exonic genome reaching at least 10× coverage. A comprehensive clinical ophthalmic examination was performed, including anterior segment imaging.

### 2.2. RNAseq Analysis

Corneal tissue was harvested from a superficial keratectomy of the proband’s brother as part of routine clinical care. Tissue was stored in RNAlater, with RNA subsequently extracted using the Qiagen RNeasy mini kit (Qiagen; Hilden, Germany). Following a quality assessment with Agilent Tapestation [17], RNA underwent Poly(A) enrichment via the NEBNext Poly(A) mRNA Magnetic Isolation Module and cDNA library generation with the xGen Broad-Range RNA Library Prep kit using xGen Normalase UDI primers with assistance from the University of Michigan Advanced Genomics Core. Afterward, paired-end sequencing was performed on an Illumina NovaSeq flow cell (San Diego, CA, USA). The resulting reads were trimmed with CutAdapt [18], checked for quality with FastQC [19], aligned to the GRCh38 reference genome with STAR v2.7.8a [20], and recorded in count matrices with RSEM v1.3.3 [21]. Using the DESeq2 package for R [22], differentially expressed gene (DEGs) analysis was performed comparing the raw sequencing counts from the proband to the raw counts of 10 published control corneal samples from individuals aged 23–46 years (median 34 years) that underwent laser photorefractive keratectomies to correct myopia [23]. As we expected a high number of false positives stemming from batch effect and difficulty with batch correction due to the small experimental sample size, we then selected only DEGs with an adjusted *p*-value of less than or equal to 1 × 10^−10^ for a PANTHER Gene-Ontology analysis [24] checking for pathway enrichment among over- and under-expressed genes. Data were visualized in R with volcano plots and heatmaps [25,26].

## 3. Results

### 3.1. Clinical Report

#### 3.1.1. Proband (II-1)

The proband (Figure 1) was born to a primiparous female with an uncomplicated prenatal course. The proband was delivered at full-term by emergency cesarean section for frank breech positioning, and her birth weight was normal. She exhibited feeding and breathing difficulty, as well as episodes of cyanosis. She was diagnosed with tracheomalacia and airway compromise with various sequelae over time, including subglottic stenosis, tonsillar hypertrophy, chronic tonsillitis, and adenoiditis.

The female proband presented to ophthalmology at age 5 with vision loss and ocular irritation. She had additional features of gross motor and speech delays but no regression or learning difficulty, bilateral sensorineural hearing loss, migraine headaches, asthma, gastroesophageal reflux, chronic cough, dysphagia, eczema, scoliosis, bowel and bladder incontinence, and eustachian tube dysfunction. At that time and subsequently, skin, nails, extremities, oral/dental, and cardiopulmonary examinations were within normal limits. No signs of ectodermal dysplasia or systemic dyskeratosis were present. MRI brain revealed mild thinning of the corpus callosum. On ophthalmic examination, the patient had corneal scarring in the left (OS) > right (OD) eye with corresponding bilateral epithelial and anterior stromal haze. Best corrected visual acuity (BCVA) was 20/50 OD and 20/125 OS. The patient underwent an initial superficial keratectomy followed by a superficial keratectomy with mitomycin C and amniotic membrane graft OS at age 5 after the recurrence of scarring. The patient’s disease course subsequently stabilized. At age 13, on ophthalmic examination, the patient had a normal retinal exam, including fundus autofluorescence. She had intraocular pressures (IOP) of 22 mm Hg OD and 26 mm Hg OS likely in the setting of a corneal scar. Corneal examination showed progressive corneal opacity with mild superior stromal haze OD and central anterior stromal haze and epithelial irregularity without defect OS (Figure 2). Focal nuclear-speckled opacities were noted in the lens OU. She had BCVA 20/25 OD and 20/100 OS.

#### 3.1.2. Proband’s Brother (II-2)

The proband’s brother had uncomplicated prenatal and newborn history and was born full-term by cesarean section with normal birth weight.

The proband’s brother presented to ophthalmology at age 8 with decreased vision OD. He had a past medical history of eustachian tube dysfunction, tonsillar hypertrophy, chronic adenotonsillitis, sleep-disordered breathing, eosinophilic esophagitis, gastroesophageal reflux, and speech and language delay. On medical examination, he additionally exhibited sensorineural hearing loss in the setting of a history of frequent ear infections. Skin, nail, extremity, and oral/dental examination were within normal limits with no signs of ectodermal dysplasia. On ophthalmic examination, his corneas exhibited bilateral epithelial and anterior stromal haze greater in OD than in OS. He had corneal scarring OU (OD > OS) without any evidence of active inflammation or corneal neovascularization. He was noted to have ocular hypertension (IOP 26 mmHg OD and 28 mmHg OS). BCVA was 20/50 OD and 20/25 OS. His visual acuity deteriorated by age 9 to 20/150 OD and 20/30 OS. He was noted to have a progression of corneal anterior stromal opacities OU with encroaching on the visual axis (Figure 2). He subsequently underwent superficial keratectomy OD, and the discarded epithelial sample was saved for RNAseq analysis. He continued to have progressive worsening of corneal opacity despite intervention, with a decline in BCVA to 20/400 OD and 20/200 OS.

#### 3.1.3. Proband’s Parents

The proband’s father had a medical history of bilateral sensorineural hearing loss status post cochlear implants, obstructive sleep apnea, degenerative disk disease, learning disability, and tracheal collapse during surgery. With respect to the hearing loss, he had moderate sloping to profound sensorineural hearing loss in the left and profound sensorineural hearing loss in the right. On ophthalmic examination, he had mild stromal opacities bilaterally that resembled the less affected eyes in both children. The proband’s mother is unaffected, and no other family members have features of early-onset sensorineural hearing loss, tracheomalacia or laryngomalacia, or vision loss/corneal opacity. Additional family members were not available for genotyping or detailed phenotyping.

### 3.2. Genetic Analysis

Chromosomal microarray, karyotype testing, and WES were performed on the proband, her brother, and parents. Peripheral blood karyotype testing was normal. Clinical WES was performed on all four family members to rule out any additional mutations that may explain the phenotype or hidden variation within specific genes in the deletion interval that could lead to homozygous loss of function. WES revealed only two potential variants segregating in the proband, her brother, and mildly affected father, neither of which were compelling (Appendix A).

Chromosomal microarray identified a paternally inherited heterozygous 1.2 Mb deletion at chromosome 13q12.11 ([hg19] chr13: 21,012,631–22,224,753) that segregates in both siblings. The genes included in the region of the 13q12.11 microdeletion are: *CRYL1*, *IFT88*, *IL17D*, *EEF1AKMT1*, *XPO4*, *LATS2*, *SAP18*, *SKA3*, *MRP63*, *MIPEPP3*, *ZDHHC20*, and *MICU20* (Figure 3 and Appendix A). Of the genes within the interval, *XPO4* and *LATS2* are constrained against loss of function, with pLI (loss of function intolerance) scores of 1 and 0.99, respectively.

A review of the literature and the DECIPHER database [13,14,15,16,27] identified no individuals with the same deletion, though several smaller and larger overlapping deletions have been described (Table 1 and Figure 3). Clinical features of these overlapping deletions that were also present in our patients include hearing impairment, laryngomalacia, global developmental delay, delayed speech and language development, and intellectual disability; individuals also exhibited atrial septal defect, cryptorchidism, microcephaly, myopia, and hypotonia. Corneal opacification was not reported in this cohort.

### 3.3. RNAseq Analysis

In order to resolve which genes within or near the deletion interval were likely contributing to pathogenesis, we sequenced RNA from a corneal epithelial sample obtained during a superficial keratectomy of the proband’s brother at age 9 years and compared the results to previously published RNAseq data from 10 healthy adult corneal epithelial samples obtained from photorefractive keratectomy for myopia [23]. Within the microdeletion, we found that the brother exhibited significantly reduced expression of *XPO4* (log_2_foldchange (log_2_fc) = −2.5, *p* = 2.88 × 10^−11^), *IFT88* (log_2_fc = −1.95, *p* = 3.74 × 10^−4^), *ZDHHC10* (log_2_fc = −1.65, *p* = 3.55 × 10^−4^), *LATS2* (log_2_fc = −1.57, *p* = 2.5 × 10^−3^), *SAP18* (log_2_fc = −1.06, *p* = 2.69 × 10^−3^), and *EEF1AKMT1* (log_2_fc = −1.26, *p* = 7.48 × 10^−3^), with all of these genes except *EEF1AKMT1* averaging moderate-to-high levels of corneal expression in controls (Figure 4). Our case sample also demonstrated significantly elevated levels of the *IL17D* transcript (log_2_fc = 3.6, *p* = 8.96 × 10^−3^), an immune effector that may be involved in corneal wound healing [28]. In terms of nearby genes—including the deafness- and cataract-associated connexins (*GJA3*, *GJB2*, and *GJB6*) [29]—we found significantly reduced expression of only *GJB6* (log_2_fc = −1.06, *p* = 3.47 × 10^−4^, Figure 4).

Global analysis further revealed a total of 4297 genes displaying significant expression differences between the case and controls, with a bias towards overexpressed genes. As this large number is likely due, in part, to batch effect, we used only those genes with an adjusted *p*-value of less than or equal to 1 × 10^−10^ for PANTHER Gene-Ontology analysis [24]. While the result showed no pathway enrichment among downregulated genes, there was a clear upregulation of genes involved with collagen metabolism and ECM formation/maintenance (Figure 5 and Appendix A).

## 4. Discussion

This report contributes to the description of 13q12.11 deletions in the literature and characterizes a unique syndrome of corneal opacification with predominant features of irregular epithelium and progressive variable anterior stromal haze with recurrence after superficial keratectomy, sensorineural hearing loss, and tracheomalacia/laryngomalacia. The corneas exhibit epithelial nummular haze and surface irregularities, subepithelial haze and deposits, and anterior stromal opacities. These features appeared to be distributed throughout the cornea (centrally/paracentrally, peripherally, temporally, and inferonasally) and progressive. The phenotype resembles a corneal scar in the absence of an inciting event, which is consistent with aberrant wound healing. While karyotype testing and WES failed to return any convincingly pathogenic variants, a chromosomal microarray revealed a 1.2 Mb deletion at chromosome 13q12.11 with trio segregation. This region demonstrates a paucity of structural variants in the general population, and most of the genes within the deletion exhibit high expression throughout the corneal epithelium. RNA-seq analysis further showed that several of these genes (*XPO4*, *IFT88*, *ZDHHC20*, *LATS2*, *SAP18*, *EFF1AKMT1*, and *GJB6*) are significantly under-expressed in the proband’s brother as compared to control samples.

Chromosome 13q12 deletions are rare; large deletions have been described and are associated with complex phenotypes of developmental delay, congenital anatomic abnormalities, and other anomalies, though ocular features have been incompletely characterized [30,31]. Smaller microdeletions have not been well characterized to date. Variable phenotypes have been described in association with 13q12 microdeletions, including sensorineural hearing impairment, intellectual disability, developmental delay, laryngomalacia, and microcephaly [16]. Reported microdeletions share several genes in common with our family’s microdeletion, most notably *LATS2, IFT88, XPO4,* and *ZDHHC20*. 

Three reports of microdeletions overlapping with that in our family have been described ranging from 1.94–2.9 Mb [13,14,15] (Figure 3). Shared features of these microdeletion syndromes with our family include developmental delay, speech delay, hearing loss, and recurrent otitis media. Additional systemic features that were non-overlapping included craniofacial dysmorphism, pectus excavatum, narrow shoulders, malformed toes, cafe-au-lait spots, hypotonia, failure to thrive, dilatation of the subarachnoid space and temporal section of both lateral ventricles, incomplete cleft palate, short external acoustic canal, small kidney cysts, clinodactyly of the fifth finger, microcephaly, scaphocephaly, and torticollis. Ophthalmic features were not fully evaluated in these prior reports but included divergent squint, hypermetropia, and high astigmatism [13,14,15].

Select connexin genes (*GJA3, GJB2*, and *GJB6*) are common to several of the aforementioned microdeletion syndromes. Connexins comprise gap junction channels and hemichannels, and mutations in human connexin genes have been linked to distinct genetic disorders including cataracts, skin disorders, and various forms of deafness [29]. The exact cause of the corneal phenotype in our reported family remains unclear. The deletion-proximal connexins (*GJA3*, *GJB2*, and *GJB6*) were initially considered genes of interest, as they have been reported as part of 13q12.11 microdeletions in the literature described above and linked to both hearing loss and cataract formation [32]; however, in our RNAseq data, only *GJB6* exhibited any significant expression differences in the affected corneal epithelium, and this effect was quite modest. None of the other genes near the deletion interval showed significant differential expression in our region-focused DEGs analysis. *GJB2* and *GJB6* have also been associated with ichthyoses, keratoderma, and ectodermal dysplasias [33]. In turn, ectodermal dysplasias are associated with progressive corneal scarring and corneal abrasions [34,35,36,37]. However, no skin or nail features consistent with ectodermal dysplasia were noted in our family despite careful examination. Together, these results suggest that connexins are unlikely to be contributing to the corneal disease phenotype.

Sensorineural hearing loss is a feature of all three family members in this report though haploinsufficiency of the genes within 13q12 microdeletion has not been associated directly with this phenotype to date. However, together with *GJB2* and *GJB6* (which are outside of the microdeletion interval), *CRYL1* constitutes part of the DFNB1 locus, which accounts for nearly half of the sensorineural hearing loss in certain populations [38]. Though many of these cases are attributed to the haploinsufficiency of *GJB2* or *GJB6* [38,39,40], it remains possible that loss of *CRYL1* contributes to disease pathogenesis. In support of this, there are several hearing-loss-associated *DFNB1* microdeletions reported in the literature [38] and in DECIPHER that include *CRYL1* but neither of the connexins.

Interestingly, there are key phenotypic overlaps between the affected family members and DECIPHER/GeneMatcher patients lacking two functional copies of *XPO4*, many of whom present with some combination of laryngomalacia and developmental delay, as well as one case of unilateral sensorineural hearing loss [16,41]. Cataracts have also been observed in *Xpo4* knockout mice [42].

Though no single gene appears to be a clear cause of the corneal phenotype, several genes are involved in pathways relevant to the observed RNAseq changes and corneal scarring. *ZDHHC20*, *IFT88*, *XPO4*, and *LATS2* all regulate the production and organization of ECM proteins [43,44,45,46,47,48,49,50,51,52,53]. These processes occur downstream of EGF and TGFβ/SMAD signaling as part of the canonical corneal wound healing response [49]. As such, mutations in associated genes have already been linked to corneal dystrophies [54].

Specifically, *ZDHHC20* is a palmitoyltransferase and serves as a negative regulator of EGF signaling via palmitoylation [48]. EGF induces the transformation of keratocytes into proto-myofibroblasts during corneal wound healing, working synergistically with TGFβ signaling to produce ECM components [49,50]. *IFT88*, an intraflagellar transport protein, is essential to stromal keratocyte organization [43]. It modulates corneal ECM through a variety of processes, including collagen production and fiber organization, regulation of Hedgehog signaling, ciliary calcium response, and protease endocytosis [44,45,46]. *Ift88* knockout mice thus develop anterior segment dysgenesis, and there is a reported case of human corneal opacification as the result of ciliopathy [43,47].

Importantly, however, *ZDHHC20* and *IFT88* both have pLI scores of 0, while *XPO4* and *LATS2* demonstrate far greater intolerance to loss of function with pLI scores of 1 and 0.99, respectively [51]. *XPO4* expression is negatively correlated with the phosphorylation and nuclear localization of the SMAD3 transcription factor, both of which are required for its role as a positive effector of TGFβ signaling [52]. LATS2 is a negative regulator of the YAP/TAZ signaling, which may positively regulate the production of and/or response to TGFβ ligands [53]. Loss of *ZDHHC20*, *IFT88*, *XPO4*, and *LATS2*, or some combination thereof, may therefore lead to the aberrant production/deposition of ECM in the anterior cornea (Figure 5). This could trigger the type of positive feedback loop often seen in cases of fibrosis and ultimately lead to progressive corneal scarring [55] (Figure 6).

We have described a syndrome with novel clinical features, including corneal opacification, sensorineural hearing loss, and tracheomalacia/laryngomalacia, and identified an associated 13q12.11 microdeletion, which may be pathogenic. The features of airway collapse and hearing impairment are common among our family, and overlapping microdeletions, as previously reported, with loss of *XPO4* and part of the *DFNB1* locus fitting most likely with these phenotypes. There has been no prior description of a corneal phenotype in association with 13q12.11 microdeletions. Given the incomplete ocular examination of these patients and the variability of phenotypic presentations within our reported family, it is possible that corneal features were overlooked in these severe syndromic conditions. Though no single gene within the identified deletion appears to be directly associated with the corneal phenotype, we have identified candidate genes involved in the regulation and production of ECM in the cornea, including *ZDHHC20, IFT88, XPO4,* and *LATS2*.

The recognition of the aforementioned constellation of features as a syndromic corneal opacification will improve the identification and clinical management of this cohort of patients. Further studies to define the genetic etiology of syndromes featuring progressive corneal opacification like this one may inform diagnosis and clinical and surgical treatment options. The cornea’s accessibility as part of the anterior segment and immune privilege makes it an appealing target for gene therapy [56]. In vitro and animal model studies have validated gene therapy in the anterior ocular segment, including anti-fibrotic treatment targeting downstream targets of TGFβ signaling. Further elucidation and characterization of the genetic etiology for this novel syndromic corneal opacification have the potential to inform genetic screening as well as identify gene targets for future clinical trials and gene therapy approaches.

## Figures and Tables

**Figure 1 genes-14-01034-f001:**
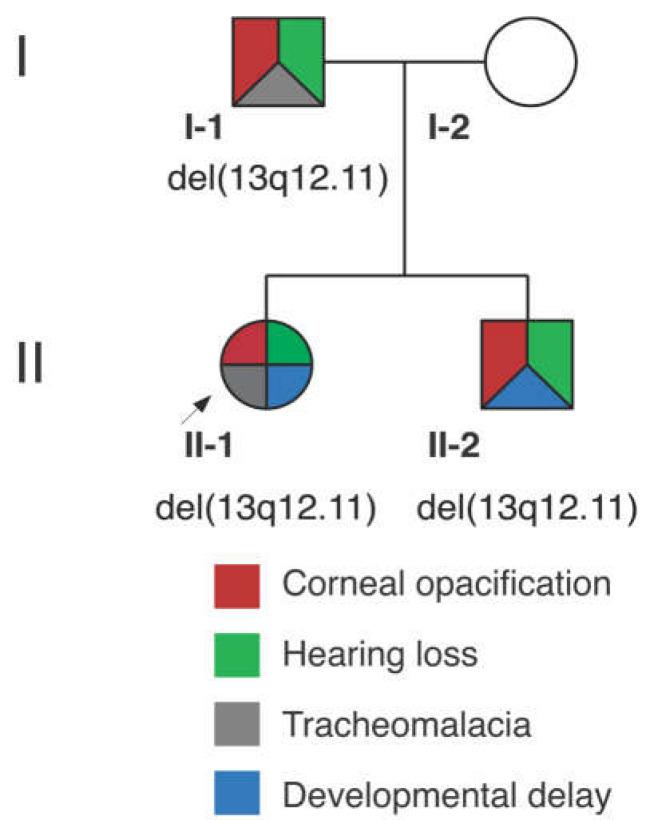
Pedigree depicting 13q12.11 deletion in affected family members and their corresponding phenotypes.

**Figure 2 genes-14-01034-f002:**
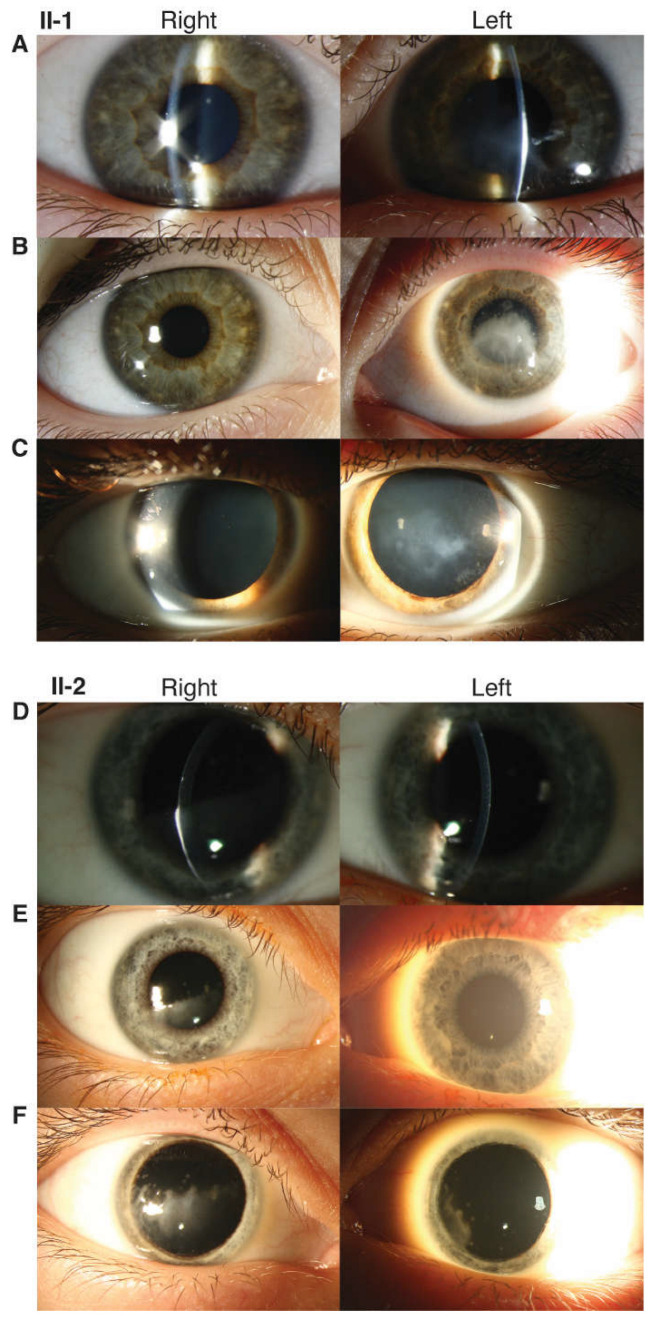
Clinical features of syndromic corneal opacification family. II-1: Slit lamp (**A**) and external (**B**) photos at presentation (age 5 years) and external photos at follow-up after superficial keratectomy (age 13 years) (**C**) II-2: Slit lamp (**D**) and external (**E**) photos at presentation (age 8 years) and external photos at follow-up after superficial keratectomy (age 9 years) (**F**). The current age of the proband (II-1) is 15, and the proband’s brother (II-2) is 12.

**Figure 3 genes-14-01034-f003:**
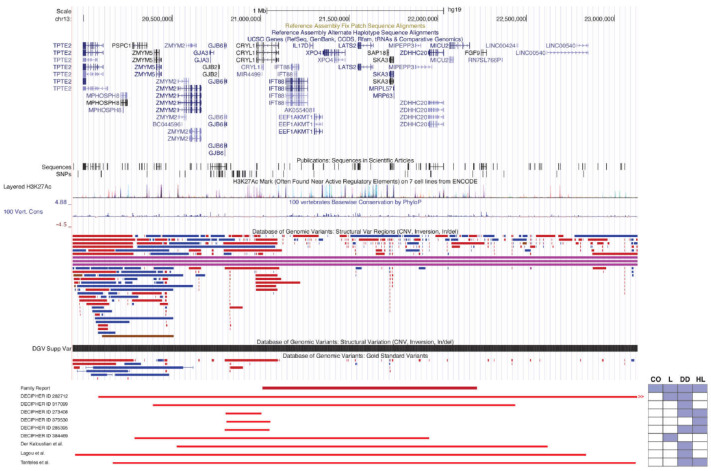
Chromosome 13q12 microdeletion interval showing genes and overlapping microdeletions. UCSC genome browser region depicting region and involved genes of family’s microdeletion as well as overlapping microdeletions identified by literature review and the DECIPHER database. Browser coordinates use the hg19 genome build. Phenotypes present for each microdeletion are highlighted [13,14,15].

**Figure 4 genes-14-01034-f004:**
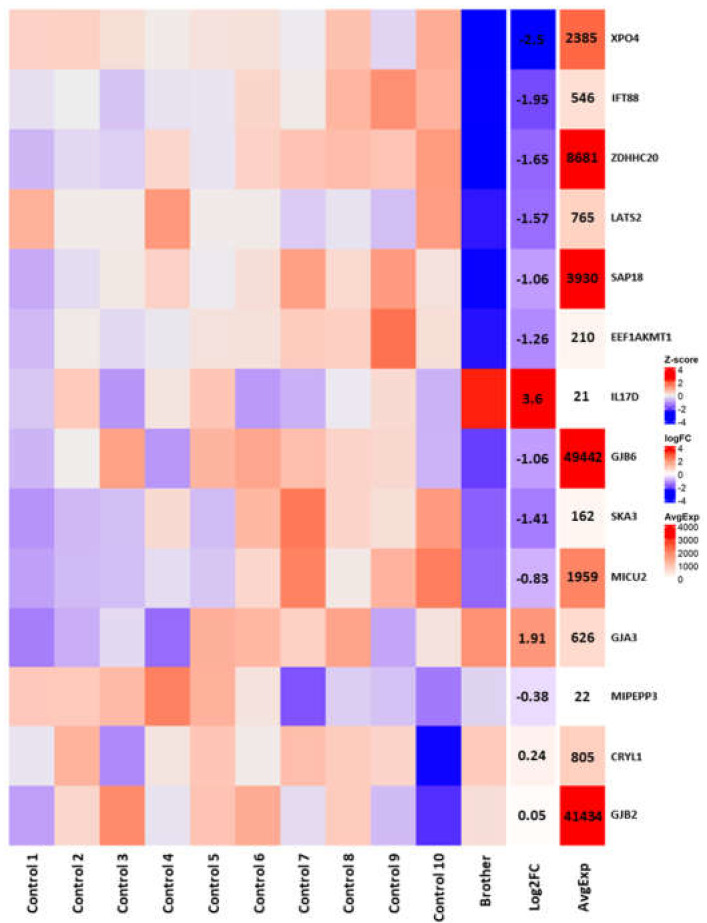
Region-focused DEGs analysis. A Heatmap showing the normalized Z-scores and log_2_ fold changes (log_2_fc) for the genes in/around the microdeletion, as well as their average corneal expression in normalized counts (AvgExp) across the samples.

**Figure 5 genes-14-01034-f005:**
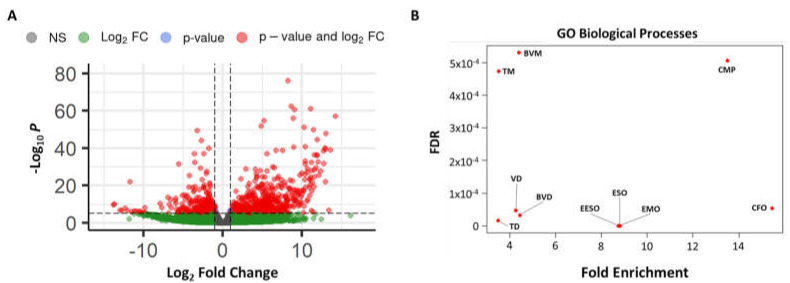
RNAseq analysis from corneal epithelium. (**A**) A volcano plot showing the DESeq2 comparison of global gene expression between a corneal epithelial sample from our case and controls (healthy, mildly myopic patients). Green is log_2_ fold change (FC) >|2| and *p*-value > 10^−6^; blue is log_2_ fold change <|2| and *p*-value < 10^−6^; red is log_2_ fold change >|2| and *p*-value < 10^−6^. (**B**) A scatter plot showing the fold enrichment of the top ten most significantly upregulated Gene Ontology (GO) processes from our case sample as determined by the lowest false discovery rate (FDR). TM, tube morphogenesis; BVM, blood vessel morphogenesis; CMP, collagen metabolic processes; TD, tube development; VD, vasculature development; BVD, blood vessel development; EESO, external encapsulating structure organization; EMO, extracellular matrix organization; CFO, collagen fibril organization.

**Figure 6 genes-14-01034-f006:**
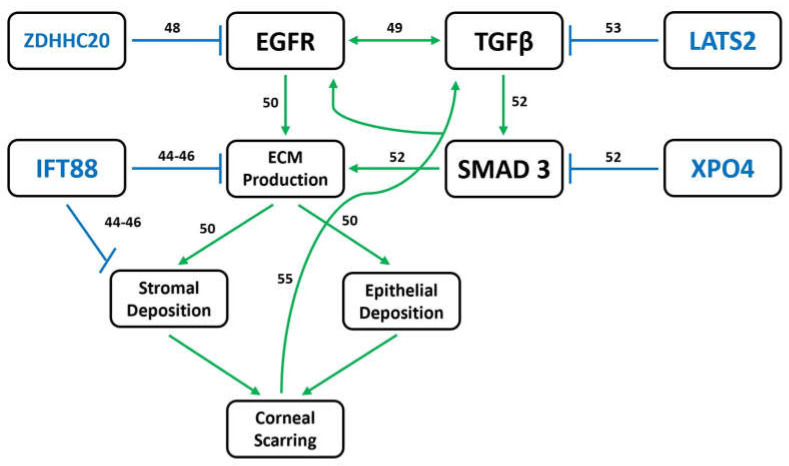
Model for dysregulated gene expression leading to corneal opacification phenotype. EGF and TGFβ signaling work synergistically to promote ECM protein formation/deposition in the cornea, a process negatively regulated by four of the genes lost in the microdeletion: *XPO4*, *LATS2*, *ZDHHC20*, and *IFT88*. *LATS2* is thought to inhibit TGFβ signaling through its role in the YAZ/TAP pathway, whereas *XPO4* inhibits the function/localization of key TGFβ effector SMAD3. Additionally, *ZDHHC20* inhibits EGF signaling via palmitoylation of EGF receptors (EGFR). *IFT88* works downstream of both pathways in regulating the ECM through its roles in collagen production/fiber organization, Hedgehog signaling, ciliary calcium response, and protease endocytosis. The loss of one or more of these genes could thus result in the inappropriate accumulation of ECM proteins in the anterior cornea, triggering a fibrotic response and resulting in a progressive haze.

**Table 1 genes-14-01034-t001:** Overlapping Variants and Phenotypes.

Variant (hg19 Coordinates)	Overlapping Phenotype	Includes *XPO4*/*CRYL1*	Reference
del 13: 20,079,051–25,514,640	Laryngomalacia; motor, language, and speech delays	*XPO4*, *CRYL1*	DECIPHER ID 282712
del 13: 20,407,295–22,453,812	Developmental delay	*XPO4*, *CRYL1*	DECIPHER ID 317099
del 13: 19,938,561–22,840,254	Developmental delay, speech delay	*XPO4*, *CRYL1*	Lagou et al. [14]
del 13: 20,174,448–23,128,904	Developmental delay affecting speech and language, recurrent otitis media, conductive hearing loss	*XPO4*, *CRYL1*	Tanteles et al. [15]
del 13: 20,521,989–22,617,211	Developmental delay	*XPO4*, *CRYL1*	Der Kaloustian et al. [13]
del 13: 20,808,367–21,001,431	Intellectual disability, sensorineural hearing impairment	*CRYL1* *	DECIPHER ID 273408
del 13: 20,808,544–21,078,913	Bilateral conductive hearing impairment	*CRYL1* *	DECIPHER ID 379530
del 13: 20,797,139–21,059,969	Intellectual disability, sensorineural hearing impairment	*CRYL1*	DECIPHER ID 285395
del 13: 20,281,273–21,945,915	Laryngomalacia, stridor	*XPO4*, *CRYL1*	DECIPHER ID 384469

* Does not include connexin genes (*GJA3*, *GJB2*, *GJB6*).

## Data Availability

All corneal RNAseq data files generated for this study are submitted to the NCBI Gene Expression Omnibus database (http://www.ncbi.nlm.nih.gov/geo) with the accession number GSE230430 (deposited 21 April 2023).

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
