# Peer review of "A Novel 13q12 Microdeletion Associated with Familial Syndromic Corneal Opacification"

_genes, 2023, doi:10.3390/genes14051034_

Round 1

Reviewer 1 Report

General comments

The manuscript reports on a two-generation family with three individuals affected by corneal opacity characterized by epithelial and anterior stroma scarring and sensorineural hearing loss in two and tracheomalacia in two. All carried a 1.2 Mb deletion at chromosome 13q12.11, with no other noteworthy co-segregating variants identified on clinical exome or chromosomal microarray. In the search for possibly disease-causing gene, the authors performed RNA-Seq analysis from an affected corneal epithelial sample which revealed downregulation of XPO4, IFT88, ZDHHC20, LATS2, SAP18, and EEF1AKMT1 within the microdeletion interval.

The results described seem to explain the sensorineural hearing loss however as for corneal findings a significant possibility remains that the molecular genetic cause is located elsewhere and thus is independent of the detected deletion. RNA-Seq from only one sample compared to RNA-Seq published in the literature as control may give incomparable results.

The methodology completely lacks details on exome sequencing and variant filtering strategy.

The discussion section is too long and should be shortened.

As corneal dystrophies are by definition non-syndromic conditions, i.e. with no systemic findings the corneal findings present in the manuscript should not be denoted as corneal dystrophy. (Definition from IC3D manuscript: In ophthalmology, corneal dystrophies have typically referred to a group of inherited disorders that are usually bilateral, symmetric, slowly progressive, and not related to environmental or systemic factors.

In general, the manuscript lacks deeper understating of corneal dystrophies. In summary, the manuscript provides some novel information but needs to be extensively rewritten.

Minor comments

Line 37 Corneal dystrophies are rare (~9/10000) abnormalities of one or several layers of the 37 cornea and have a genetic or hereditary epigenetic origin [1].

 Correct to 10,000

 Line 38 They present with variably shaped corneal opacities with varying effects on visual acuity

Variably-varying.. doesn‘t read well, I would suggest rewording

Line 45 the international committee for classification of corneal dystrophies

 This is usually written in capital letters.

Line 45 While some corneal dystrophies have defined genetic etiologies, (i.e. congenital hereditary endothelial dystrophy, posterior polymorphous corneal dystrophy, and Fuch’s endothelial corneal dystrophy) [4], for most corneal dystrophies, a genotype-phenotype correlation has not been demonstrated [5].

 Fuch’s – correct is either Fuchs or Fuchs'

Genetic aetiologies have been defined for most corneal dystrophies and for most there is good genotype-phenotype correlation.

Line 52 Many anterior corneal dystrophies, i.e. granular and lattice dystrophy, result from dominant variants in TGFB1 [6]. Within the cornea, the TGFB1

It is not TGFB1 but TGFBI

Line 58 While most corneal dystrophies 58 are isolated to the cornea, some have systemic features, including Schnyder corneal dystrophy and Meretoja Syndrome [2]. 60

Meretoja syndrome is not counted among conrneal dystrphies- see IC3D version 2 (The so-called lattice corneal dystrophy type 2 (LCD2) is a misnomer,  in fact comprising systemic amyloidosis plus corneal lattice lines and should be termed familial amyloidosis, Finnish type, or gelsolin type. Eponymously, it is known as Meretoja syndrome (Figs. 11A, B).

Line 71 Chromosomal microarray, karyotype testing, and clinical whole exome sequencing (WES) was performed on the proband, her brother, her father, and her unaffected mother.

More details need to be provided- library kit, sequencing platform, tools used for alignment, and human genome version. Also variant calling and filtering.

Abbreviations OS and OD have not been introduced (while ophthalmologists know what it means laboratory staff may not).

Figure 1 is very untidy it should be splitted into two or three. Also I miss corneal findings of the father. Why is proband (corneal findings) shown after the sibling?

Line 234 a whole exome se- 234 quencing failed

Abbreviation WES has been introduced

Line 307 Moreover, a significant portion of the known corneal dystrophy genes fall into related 306 categories [52]. These include TGFβ-regulated transcription factors TCF4 and ZEB1 307 [53,54]; TGFβ antagonists OVOL2 and GRHL2 [55,56]; and ECM constituent/organizational protein encoding genes TGFBI, DCN, EPYC, and LUM [7,52,57].

The mechanisms involved in corneal dystrophies are different, not so simple as presented.

Please delete.

 Line 341 We have described a syndromic corneal dystrophy with novel clinical features

As already stated above there is no syndromic corneal dystrophy.

Author Response

We thank the reviewers for their helpful comments and have provided a point-by-point response. 

Reviewer #1 

The manuscript reports on a two-generation family with three individuals affected by corneal opacity characterized by epithelial and anterior stroma scarring and sensorineural hearing loss in two and tracheomalacia in two. All carried a 1.2 Mb deletion at chromosome 13q12.11, with no other noteworthy co-segregating variants identified on clinical exome or chromosomal microarray. In the search for possibly disease-causing gene, the authors performed RNA-Seq analysis from an affected corneal epithelial sample which revealed downregulation of XPO4, IFT88, ZDHHC20, LATS2, SAP18, and EEF1AKMT1 within the microdeletion interval. 

The results described seem to explain the sensorineural hearing loss however as for corneal findings a significant possibility remains that the molecular genetic cause is located elsewhere and thus is independent of the detected deletion. RNA-Seq from only one sample compared to RNA-Seq published in the literature as control may give incomparable results.  

We acknowledge the limitations of RNAseq analysis from a single sample and the batch effects from comparison to published data.  Nonetheless, our analysis is valuable and given the rarity of this condition and the inability to access additional patient samples, having another replicate is not feasible.  

Extensive sequencing and copy-number variant analysis was done to rule out other causes, including whole exome sequencing and chromosomal microarray, respectively.  We have included information about these pertinent negative results in the text.   

The methodology completely lacks details on exome sequencing and variant filtering strategy. 

The exome sequencing details have been added in the methods section.  This was done with the standard GeneDx pipeline for familial exome sequencing using the commercial laboratory. We have included details on this in the Methods section:  

(Line 107): “Chromosomal microarray and karyotype testing were performed on the proband, with confirmatory testing on her brother, her father, and her unaffected mother using standard clinical based testing, with Nimblechip HG18 385,000 probe array (Roche, Madison, WI). DNA samples were also collected from these 4 individuals with OraCollect buccal swabs and submitted for clinical whole exome sequencing (WES) using the XomeDxPlus platform (GeneDx, Stamford, CT, USA) and processed using the standard GeneDx clinical pipeline, including the GeneDx proprietary capture kit (NGS-CNV), paired-end sequencing on the Illumina platform, alignment with genome build GRCh37/UCSC hg 19 and XomeAnalyzer for variant (including CNVs of greater than 3 exons) calling and filtering. Mean depth of coverage was 121x, with 98.9% of the exonic genome reaching at least 10x coverage.” 

The discussion section is too long and should be shortened. 

We have removed the suggested areas of the discussion as below and shortened several areas of the discussion as requested.   

As corneal dystrophies are by definition non-syndromic conditions, i.e. with no systemic findings the corneal findings present in the manuscript should not be denoted as corneal dystrophy. (Definition from IC3D manuscript: In ophthalmology, corneal dystrophies have typically referred to a group of inherited disorders that are usually bilateral, symmetric, slowly progressive, and not related to environmental or systemic factors. 

We have classified the familial disorder as a syndrome associated with corneal opacification and do not refer to it as a corneal dystrophy.  We have still included a small paragraph on TGFBI dystrophies and other corneal dystrophies given that dysregulation of TGFBI pathway genes is seen in the corneal epithelial sample from the proband’s brother.   

In general, the manuscript lacks deeper understating of corneal dystrophies. In summary, the manuscript provides some novel information but needs to be extensively rewritten. 

We have revised the introduction to cover syndromes associated with corneal opacification and shorted the section on corneal dystrophies in the discussion and introduction.   

Minor comments 

Line 37 Corneal dystrophies are rare (~9/10000) abnormalities of one or several layers of the 37 cornea and have a genetic or hereditary epigenetic origin [1]. 

 We have corrected to 10,000 in line 37.   

 Line 38 They present with variably shaped corneal opacities with varying effects on visual acuity 

Variably-varying.. doesn‘t read well, I would suggest rewording 

We replaced “varying” with “a spectrum of effects.” 

Line 45 the international committee for classification of corneal dystrophies 

 This is usually written in capital letters. 

We have changed this to capital letters. 

Line 45 While some corneal dystrophies have defined genetic etiologies, (i.e. congenital hereditary endothelial dystrophy, posterior polymorphous corneal dystrophy, and Fuch’s endothelial corneal dystrophy) [4], for most corneal dystrophies, a genotype-phenotype correlation has not been demonstrated [5]. 

 Fuch’s – correct is either Fuchs or Fuchs' 

Genetic aetiologies have been defined for most corneal dystrophies and for most there is good genotype-phenotype correlation. 

We have modified this statement to reflect that many corneal dystrophies have defined genetic etiologies, and rephrased that for some, I.e. Lisch corneal dystrophy, the etiology has not been defined.  We have revised this statement to “While many corneal dystrophies have defined genetic etiologies, (i.e. congenital hereditary endothelial dystrophy, posterior polymorphous corneal dystrophy, and Fuchs’ endothelial corneal dystrophy) [4], for other corneal dystrophies, the genetic etiology has not been identified [5].” 

Line 52 Many anterior corneal dystrophies, i.e. granular and lattice dystrophy, result from dominant variants in TGFB1 [6]. Within the cornea, the TGFB1 

It is not TGFB1 but TGFBI 

We have changed TGFB1 to TGFBI in all instances and appreciate the reviewer’s attention to this oversight.   

Line 58 While most corneal dystrophies 58 are isolated to the cornea, some have systemic features, including Schnyder corneal dystrophy and Meretoja Syndrome [2]. 60 

Meretoja syndrome is not counted among conrneal dystrphies- see IC3D version 2 (The so-called lattice corneal dystrophy type 2 (LCD2) is a misnomer,  in fact comprising systemic amyloidosis plus corneal lattice lines and should be termed familial amyloidosis, Finnish type, or gelsolin type. Eponymously, it is known as Meretoja syndrome (Figs. 11A, B). 

We appreciate these comments for using the IC3D classification for corneal dystrophies and have referred to syndromes associated with progressive corneal opacification and categorized our newly identified syndrome in this context.  We have thus revised the introduction to reflect the syndromes associated with corneal opacification separately, with focus on ectodermal dysplasia, limbal stem cell deficiency, and other corneal opacities (I.e. familial amyloidosis):  

Corneal involvement is associated with a variety of multisystemic disorders. For example, gelsolin amyloidosis is a multisystem disorder clinically associated with corneal amyloidosis that resembles lattice corneal dystrophy [8]. Keratopathy is also present in inborn errors of metabolism including mucopolysaccharidoses, mucolipidoses, cystinosis, tyrosinaemia type II, Wilson disease, Fabry disease, Lecithin Cholesterol Acyltransferase Deficiency (LCAT)-related metabolic disease, and Tangier disease [8]. Corneal opacification is a feature of various systemic conditions including ectodermal dysplasias, of which hundreds of clinically defined subtypes exist [8]. Ectodermal dysplasias affect ectoderm-derived ocular structures including the corneal epithelium. For instance, ectrodactyly-ectodermal-dysplasia clefting syndrome includes ocular features of limbal stem cell deficiency, corneal scarring, and ocular inflammation. Keratitis-ichythosis-deafness syndrome can include neovascularization, recurrent corneal epithelial defects, and limbal stem cell deficiency [8]. Limbal stem cell deficiency is a reported feature of various genetic disorders including PAX6-related aniridia, ectrodactyly-ectodermal-dysplasia-clefting syndrome, keratitis-ichythosis-deafness syndrome, xeroderma pigmentosum, and Turner syndrome.” 

Line 71 Chromosomal microarray, karyotype testing, and clinical whole exome sequencing (WES) was performed on the proband, her brother, her father, and her unaffected mother. 

More details need to be provided- library kit, sequencing platform, tools used for alignment, and human genome version. Also variant calling and filtering. 

We have included more details on variant calling and filtering and exome pipeline analysis – these were all done with a standard clinical pipeline at GeneDx. These are noted above.   

Abbreviations OS and OD have not been introduced (while ophthalmologists know what it means laboratory staff may not). 

We have added OS – left, OD – right in the first mention of these terms.  

Line 213: “On ophthalmic examination, the patient had corneal scarring in the left (OS) > right (OD) eye with corresponding bilateral epithelial and anterior stromal haze” 

Figure 1 is very untidy it should be splitted into two or three. Also I miss corneal findings of the father. Why is proband (corneal findings) shown after the sibling? 

We have split Figure 1 into two figures – with the pedigree as a separate figure and the clinical findings as a separate figure.  The proband (II-2 is listed first in the clinical findings figure).  We have enlarged Figure 2 so that details are more visible.  While a clinical exam for the father was available, we unfortunately were not able to get slit lamp photos of his corneas.  

Line 234 a whole exome se- 234 quencing failed 

Abbreviation WES has been introduced 

We have changed the abbreviated term WES.   

Line 307 Moreover, a significant portion of the known corneal dystrophy genes fall into related 306 categories [52]. These include TGFβ-regulated transcription factors TCF4 and ZEB1 307 [53,54]; TGFβ antagonists OVOL2 and GRHL2 [55,56]; and ECM constituent/organizational protein encoding genes TGFBI, DCN, EPYC, and LUM [7,52,57]. 

The mechanisms involved in corneal dystrophies are different, not so simple as presented. 

Please delete. 

We have deleted these statements in the discussion.   

 Line 341 We have described a syndromic corneal dystrophy with novel clinical features 

As already stated above there is no syndromic corneal dystrophy. 

We have modified the terminology to classify this disorder as a syndrome featuring corneal opacification.   

Reviewer 2 Report

Serpen and colleagues presented a family with microdeletion presenting with corneal dystrophy, sensorineural hearing loss, tracheomalacia, some developmental delay and learning disability. They carried out RNAseq of the corneal tissue of the affected brother and identified decrease in expression of several genes XPO4, IFT88, ZDHHC20, LATS2, SAP18, and EEF1AKMT1. They compared their findings to other microdeletion reports in the region. The study is comprehensive, however the presentation in parts is difficult to follow especially comparing the phenotype of other microdeletions in the region and could benefit from a more simplified illustration or table.

Comments:

-          The current ages of the proband and family members were not listed. It is not clear at what age the superficial keratectomy was performed for the brother. If at much younger age than the controls (23-46yrs) inclusion of additional age matched controls will be helpful to exclude the effect of age as variable. Listing the median age and range of the controls will be important.

-          If the proband and her brother are still children, did they assent to the study?

-          The authors indicate that no other family members have ocular or auditory symptoms but were the authors able to examine and document that or is it just by report. Is the father denovo? Have the grandparents on the father side been tested? Expanding the pedigree especially on the father side is needed.

-          Several genes have been associated with corneal dystrophy (PMID: 31301286). Since other microdeletion in the same region with some larger than the one identified in this family don’t present with ocular manifestations a possible explanation is that a variant in other gene outside the microdeletion modify the phenotype. Were there any variants in genes associated with corneal dystrophy identified in this family? The authors list only few of the genes associated with corneal dystrophy, but a more comprehensive list of the genes assessed/excluded will be helpful.

-          The resolution of Figure 2 is low. It shows the size of the other microdeletions but doesn’t capture the phenotypes. Table 1 summarizes the other microdeletion, include chromosomal location and phenotype but it doesn’t include the references for each and only include two genes. A visual representation that includes all the genes in the region impacted by each microdeletion and the associated phenotype will be very helpful. A simplified drawing of the other microdeletion or a heatmap format that summarizes the genetic/phenotypic correlation of each microdeletion is needed.

-          A summary table of the phenotypes associated with each gene in the microdeletion region would be helpful.

Author Response

We thank the reviewers for their helpful comments and have provided a point-by-point response. 

Reviewer #2 

Serpen and colleagues presented a family with microdeletion presenting with corneal dystrophy, sensorineural hearing loss, tracheomalacia, some developmental delay and learning disability. They carried out RNAseq of the corneal tissue of the affected brother and identified decrease in expression of several genes XPO4, IFT88, ZDHHC20, LATS2, SAP18, and EEF1AKMT1. They compared their findings to other microdeletion reports in the region. The study is comprehensive, however the presentation in parts is difficult to follow especially comparing the phenotype of other microdeletions in the region and could benefit from a more simplified illustration or table. 

We appreciated the positive comments. We have modified the microdeletion figure to include a side portion that graphically outlines the phenotypic features of the microdeletion.   

Comments: 

-          The current ages of the proband and family members were not listed. It is not clear at what age the superficial keratectomy was performed for the brother. If at much younger age than the controls (23-46yrs) inclusion of additional age matched controls will be helpful to exclude the effect of age as variable. Listing the median age and range of the controls will be important. 

We have included ages – in the figures – and added current ages at last follow-up.  Given that these are human samples and that healthy patients undergoing photorefractive keratectomy surgery must be adults, it is infeasible to get perfectly age matched controls.  The published controls remain the best feasible cohort for comparison.  We acknowledge the limitations of our comparison in our manuscript.   

-          If the proband and her brother are still children, did they assent to the study? 

The proband did provide assent for the study. The probands brother was too young given his developmental status to provide meaningful assent at the time of consent, but had no objections to participating in the study.   

-          The authors indicate that no other family members have ocular or auditory symptoms but were the authors able to examine and document that or is it just by report. Is the father denovo? Have the grandparents on the father side been tested? Expanding the pedigree especially on the father side is needed. 

We unfortunately do not have access to genotypes on the father’s side.  For the phenotypic information, we have collected relevant information for the extended pedigree on the father’s side. The paternal grandfather did develop hearing loss from occupational exposure, so we qualified the statement about auditory symptoms to reflect no childhood hearing loss. A paternal uncle had learning difficulty, but his genotype is unclear.  We are unable to determine whether the father’s genotype was de novo as we unfortunately have no access to grandparent or sibling DNA. Given the lack of genotypic information and that the phenotypic information is all by report (unknown ocular status), we opted to keep the nuclear family, which was both well genotyped and phenotyped and omit the remaining family members that may be more distracting and not add value to the report.    

-          Several genes have been associated with corneal dystrophy (PMID: 31301286). Since other microdeletion in the same region with some larger than the one identified in this family don’t present with ocular manifestations a possible explanation is that a variant in other gene outside the microdeletion modify the phenotype. Were there any variants in genes associated with corneal dystrophy identified in this family? The authors list only few of the genes associated with corneal dystrophy, but a more comprehensive list of the genes assessed/excluded will be helpful. 

The family underwent whole exome sequencing with good coverage, so other coding variants and small CNVs in epithelial and stromal corneal dystrophy genes were effectively excluded.  There was a LOXLHD1 variant found, but this is only associated with Fuchs endothelial dystrophy which is not relevant to the current corneal phenotypes of the patients.   

-          The resolution of Figure 2 is low.  

      We have increased the size and resolution of the microdeletion figure Figure 2 (new Figure 3) and modified it to be landscape and bigger.   

It shows the size of the other microdeletions but doesn’t capture the phenotypes.  

We have included a side panel in new Figure 3 to capture the phenotypic features of the overlapping microdeletions, with more details, such as the precise breakpoints, provided in Table 1.    

      Table 1 summarizes the other microdeletion, include chromosomal location and phenotype but it doesn’t include the references for each and only include two genes.  

      References to each of the microdeletions are now included in Table 1.   

A visual representation that includes all the genes in the region impacted by each microdeletion and the associated phenotype will be very helpful. A simplified drawing of the other microdeletion or a heatmap format that summarizes the genetic/phenotypic correlation of each microdeletion is needed. 

       As above a small map of the key phenotypic features (cornea/tracheo or laryngeomalacia/hearing loss/developmental delay) for each microdeletion have been mapped onto new Figure 3.   

-          A summary table of the phenotypes associated with each gene in the microdeletion region would be helpful. 

A new summary table of the phenotypes associated with each gene in the microdeletion interval has been included a Supplemental Table 2.

Round 2

Reviewer 1 Report

Minor comments

Chromosomal microarray, karyotype testing, and clinical whole exome sequencing 95 (WES) was performed on the proband, her brother, her father, and her unaffected mother. 96 and karyotype testing were performed on the proband, with confirmatory testing on her 97 brother, her father, and her unaffected mother using standard clinical based testing with 98 the Nimblechip HG18 385,000 probe array (Roche, Madison, WI). DNA

 There is no need to repeat HER 3x, ale are you sure on (and not in) is correct?

Line 58 Mutated TGFBI protein appears to specifically accumulate in the cornea versus other tissues in patients with TGFBI corneal dystrophies, and corneal deposits in TGFBI-associated corneal dystrophies localize across layers of the cornea including the Bowman layer and the corneal stroma [7].

The sentence doesn’t read well- e.g. TGFBI corneal dystrophies, and corneal deposits in TGFBI-associated corneal dystrophies. I would suggest to reword.

Current age of proband 186 (II-2) is 15, and proband’s brother (II-1) is 12.

As the proband is older than her brother she should be the first form generation II in the pedigree figure. See for example:

https://medicine.uiowa.edu/humangenetics/resources/how-draw-pedigree/instructions-how-draw-pedigree

In addition, the pedigree figure should show the numbering (i.e. II-1, II-2) as well.

Figure 3: Region-focused DEGs analysis. A Heatmap [28] showing the normalized Z-scores and 238 log2 fold changes (Log2FC) for the genes in/around the microdeletion, as well as their average cor-239 neal expression in normalized counts (AvgExp) across the samples.

References in Figure legends are not usually present (this comment applies to all figure legends), please check with the journal guidelines and other previously published manuscripts.

Though no single gene appears to be a clear cause of the corneal dystrophy phenotype,

Corneal dystrophy is by definition non-syndromic so I would recommend the replace it with the term corneal phenotype

Author Response

We thank the reviewers for their comments, which we have addressed below.

Chromosomal microarray, karyotype testing, and clinical whole exome sequencing 95 (WES) was performed on the proband, her brother, her father, and her unaffected mother. 96 and karyotype testing were performed on the proband, with confirmatory testing on her 97 brother, her father, and her unaffected mother using standard clinical based testing with 98 the Nimblechip HG18 385,000 probe array (Roche, Madison, WI). DNA

 There is no need to repeat HER 3x, ale are you sure on (and not in) is correct?

We rewritten this sentence to remove her several times: “... the proband, her brother, father, and unaffected mother”.  “on” is correct for this usage as testing is performed on someone.    

Line 58 Mutated TGFBI protein appears to specifically accumulate in the cornea versus other tissues in patients with TGFBI corneal dystrophies, and corneal deposits in TGFBI-associated corneal dystrophies localize across layers of the cornea including the Bowman layer and the corneal stroma [7].

The sentence doesn’t read well- e.g. TGFBI corneal dystrophies, and corneal deposits in TGFBI-associated corneal dystrophies. I would suggest to reword.

This has been revised into two separate sentences to read more smoothly. Additionally, the term “TGFBI corneal dystrophies” was revised to “TGFBI-associated corneal dystrophies” for consistency: “Mutated TGFBI protein appears to specifically accumulate in the cornea versus other tissues in patients with TGFBI-associated corneal dystrophies. In these dystrophies, corneal deposits localize across layers of the cornea including the Bowman layer and the corneal stroma [7].”

Current age of proband 186 (II-2) is 15, and proband’s brother (II-1) is 12.

As the proband is older than her brother she should be the first form generation II in the pedigree figure. See for example:

https://medicine.uiowa.edu/humangenetics/resources/how-draw-pedigree/instructions-how-draw-pedigree

In addition, the pedigree figure should show the numbering (i.e. II-1, II-2) as well.

The pedigree has been revised to include the proband first in generation II and to include numbering for all individuals in the pedigree. We have also included the developmental delay phenotype in the pedigree to be more consistent with the subsequent figure showing the regional analysis.

Figure 3Region-focused DEGs analysis. A Heatmap [28] showing the normalized Z-scores and 238 log2 fold changes (Log2FC) for the genes in/around the microdeletion, as well as their average cor-239 neal expression in normalized counts (AvgExp) across the samples.

References in Figure legends are not usually present (this comment applies to all figure legends), please check with the journal guidelines and other previously published manuscripts.

 References have been removed from the figure legends and included in the body of the manuscript and updated.

Though no single gene appears to be a clear cause of the corneal dystrophy phenotype,

Corneal dystrophy is by definition non-syndromic so I would recommend the replace it with the term corneal phenotype

“Corneal dystrophy phenotype” was changed to “corneal phenotype.”